# Query in Your Tongue: Reinforce Large Language Models with Retrievers for Cross-lingual Search Generative Experience

## ABSTRACT

In the contemporary digital landscape, search engines play an invaluable role in information access, yet they often face challenges in Cross-Lingual Information Retrieval (CLIR). Though attempts are made to improve CLIR, current methods still leave users grappling with issues such as misplaced named entities and lost cultural context when querying in non-native languages. While some advances have been made using Neural Machine Translation models and cross-lingual representation, these are not without limitations. Enter the paradigm shift brought about by Large Language Models (LLMs), which have transformed search engines from simple retrievers to generators of contextually relevant information. This paper introduces the Multilingual Information Model for Intelligent Retrieval (MIMIR). Built on the power of LLMs, MIMIR directly responds in the language of the user's query, reducing the need for post-search translations. Our model's architecture encompasses a dual-module system: a retriever for searching multilingual documents and a responder for crafting answers in the user's desired language. Through a unique unified training framework, with the retriever serving as a reward model supervising the responder, and in turn, the responder producing synthetic data to refine the retriever's proficiency, MIMIR's retriever and responder iteratively enhance each other. Performance evaluations via CLEF and MKQA benchmarks reveal MIMIR's superiority over existing models, effectively addressing traditional CLIR challenges.

## CCS CONCEPTS

• **Information systems → Web search engines**.

## KEYWORDS

Large Language Models, Search Generative Experience, Cross-lingual Information Retrieval

**ACM Reference Format:**
Anonymous Author(s). 2018. Query in Your Tongue: Reinforce Large Language Models with Retrievers for Cross-lingual Search Generative Experience. In *Proceedings of Make sure to enter the correct conference title from your rights confirmation emai (Conference acronym 'XX).* ACM, New York, NY, USA, 10 pages. https://doi.org/XXXXXXX.XXXXXXX

## 1 INTRODUCTION

In an age characterized by the relentless pursuit of information, the role of search engines in our daily lives cannot be overstated. Search engines have become indispensable tools for accessing a vast repository of knowledge, connecting individuals with an ever-expanding digital universe. Yet, despite their ubiquity and utility, a significant limitation persists within the current landscape of search engines: their predominant focus on information retrieval within the confines of a single language. While this approach proves effective for users conducting searches in their native tongue, it often falls short in accommodating the diverse linguistic preferences and globalized communication patterns of today's internet users. It is common for individuals to find themselves unable to locate desired information when expressing their queries in their native language, only to discover that altering their search language opens a doorway to a wealth of relevant content. This conspicuous disparity highlights a critical deficiency in the realm of contemporary search engines—their inherent incapacity for Cross-Lingual Information Retrieval (CLIR).

Given the conspicuous underperformance of contemporary search engines in the realm of CLIR, researchers have made efforts to enhance the CLIR abilities with Neural Machine Translation (NMT) models or cross-lingual representation models. While these research endeavors have undoubtedly contributed to bolstering cross-lingual transferability in information retrievers, the practical application of CLIR remains severely constrained. A pivotal challenge arises when users are presented with retrieved results in languages they do not comprehend. To address this issue, existing methods have incorporated a translation model in the post-retrieval phase, aiming to translate the results into the user's native language, thereby facilitating comprehension. However, CLIR still introduces the following challenges: (1) **Named Entity Recognition (NER) Issues**: Proper nouns, especially names of places or people, might not translate directly or can get misrepresented. (2) **Cultural Topic Context Loss**: Some terms or concepts are deeply rooted in cultural context, and a straightforward translation can lose this context.

In recent years, the landscape of search engines has witnessed a transformative evolution with the advent of Large Language Models (LLMs). These models have ushered in a paradigm shift, propelling search engines beyond mere information retrieval into the realm of Search Generative Experiences (SGE). Unlike traditional search engines, which primarily return lists of matching documents, LLMs are capable of directly providing accurate and contextually relevant answers to user queries. This advancement has significantly improved the user experience, enabling more precise and efficient access to information. In this research, we try to harness the capabilities of the search generative experience to augment the practicality and utility of CLIR and emphasize the importance of ensuring that the language of the generated answer remains consistent with the language used in the user's query. By combining with LLMs, we

believe SGE can show the following advantages: (1) With extensive training data, Llms have seen numerous named entities across different contexts and languages and can recognize and correctly handle entities. (2) The accommodation of the input context length is large, which gives Llms a broader understanding of the topic and cultural contexts.

In this paper, we introduce the *Multilingual Information Model for Intelligent Retrieval* (Mimir). It enables the direct generation of responses in the user's query language, capturing and aligning with their intent, thus eliminating the need for post-retrieval translation models. Our method consists of two main modules: a retriever, which searches multilingual documents aligning with the user's query, and a responder, crafting responses matching the user's language based on the retrieved documents. To improve performance, we devised an **unsupervised unified training framework**. In responder fine-tuning, we use the retriever as a reward model, enhancing the language model's cross-lingual transferability. Conversely, for retriever refinement, we use synthetic data from the responder, boosting its performance through augmented supervision signals. These training tasks are iterative, each improving the other.

We designed experiments to assess the accuracy of our retriever and how our Llm's generated results match the user's query, using CLEF [6] and MKQA [32]. Results from these benchmarks show Mimir surpasses state-of-the-art performance against strong baselines. Further tests on entity recognition and topic translation consistency show Mimir's advantage over traditional post-retrieval translation methods.

## 2 RELATED WORK

### 2.1 Cross-lingual Information Retrieval

One line of works has tried CLIR with the help of Translation models [29, 31, 43, 53, 59]. They apply translation models to translate the multilingual queries into English or translate the retrieval document back to user's language. Researches [8, 58, 60, 62] on CLIR have been long researched through a long time, given the perspective of XLM-R and m-BERT [23, 51] with different kinds of improvement on Cross-lingual representations [13, 54–56]. Many researches [18, 19, 21, 25, 26, 40] have tried to distill the knowledge of information retrieval from monolingual model to a multilingual architecture. Methods such as code-switching [12, 27, 49], query generation [3, 40, 64] or sequential sentence relation [28, 30, 61] are also applied in the context of CLIR. VMSST [1, 52] has tried the disentanglement method, a variational generative model to separate semantic information. Another popular technique for CLIR is contrastive learning [14, 20, 46, 52, 63], researchers [17, 60, 62] have undertaken extensive efforts to enhance this crucial capability. The pursuit of improved CLIR abilities has largely converged on two primary technique routines. The first approach involves the utilization of knowledge distillation [18, 19, 25, 40], a method that commonly employs a monolingual retrieval teacher model to impart its expertise to a multilingual student architecture. Concurrently, contrastive learning [14, 20, 46, 52] has gained widespread adoption within the CLIR field due to its remarkable proficiency in aligning sentence embeddings that share similar semantics. Some

[19] have designed cross-lingual soft prompts to improve cross-lingual information retrieval. Now, CLIR has also been taken as a tool to further improve the performance of other kinds of work, such as fact-checking. [15].

### 2.2 Large Language Models for Search

The emergence of Llms, typified by ChatGPT [1], has revolutionized natural language processing due to their remarkable language understanding, generation, generalization, and reasoning abilities. Recent research has sought to leverage Llms to improve IR systems. Given the rapid evolution of this research trajectory, the confluence of LLMs and IR systems has emerged in different aspects, including crucial aspects such as query rewriters [10, 33–35, 42, 45, 50], retrievers [7, 47, 66], rerankers [4, 24, 39], and readers [22, 48]. In this paper, we focus on leveraging Llms to alleviate CLIR problems.

## 3 METHODOLOGY

We introduce the *Multilingual Information Model for Intelligent Retrieval* (Mimir). At the heart of Mimir are two pivotal models: the *Retriever* ($R_t$) and the *Responder* ($R_p$), as illustrated in Figure 1. To enhance Mimir's precision and robustness, unsupervised query augmentation is employed during its training phase. When provided with a document $D_y$ in language $y$, the *Responder* ($R_p$) generates two sets: a positive query set $Q^+$, consisting of diverse queries that align with the content of $D_y$, and a negative query set $Q^-$, containing queries closely related, yet not answerable solely using $D_y$. Leveraging contrastive learning, the *Retriever* ($R_t$) is then fine-tuned using both query sets. In parallel, the *Responder* ($R_p$) undergoes refinement via a reward signal $\mathcal{R}_{<Q,D_y>}$, sourced from the *Retriever* ($R_t$), and harnessing reinforcement learning mechanisms. A comprehensive breakdown of these modules follows in this section.

### 3.1 Synthetic Query Generation Using Responder

The quality of the synthetic queries plays a central role in Mimir's training paradigm. Presented with the document $D_y$, the *Responder* ($R_p$) generates two kinds of synthetic queries: positive queries, which resonate with the document's content, and negative queries, which, while closely related (often touching upon the same topic or entities) cannot be satisfactorily answered using only the document in question. To direct the *Responder*'s query generation process, we employ the following prompts:

- **Positive:** *"Given the content of the document: [document]. Based on the content and essence of the provided document, generate a user-like query in [target_language]."*
- **Negative:** *"Given the content of the document: [document]. Devise a query in [target_language] that, while related, cannot be fully addressed by the provided document's content."*

Substituting appropriate values for *[target_language]*, the *Responder* creates $N^+$ distinct positive queries and $N^-$ related yet unanswerable negative queries in different target languages. This nuanced approach to multilingual query generation captures the

---

[1]https://chat.openai.com/

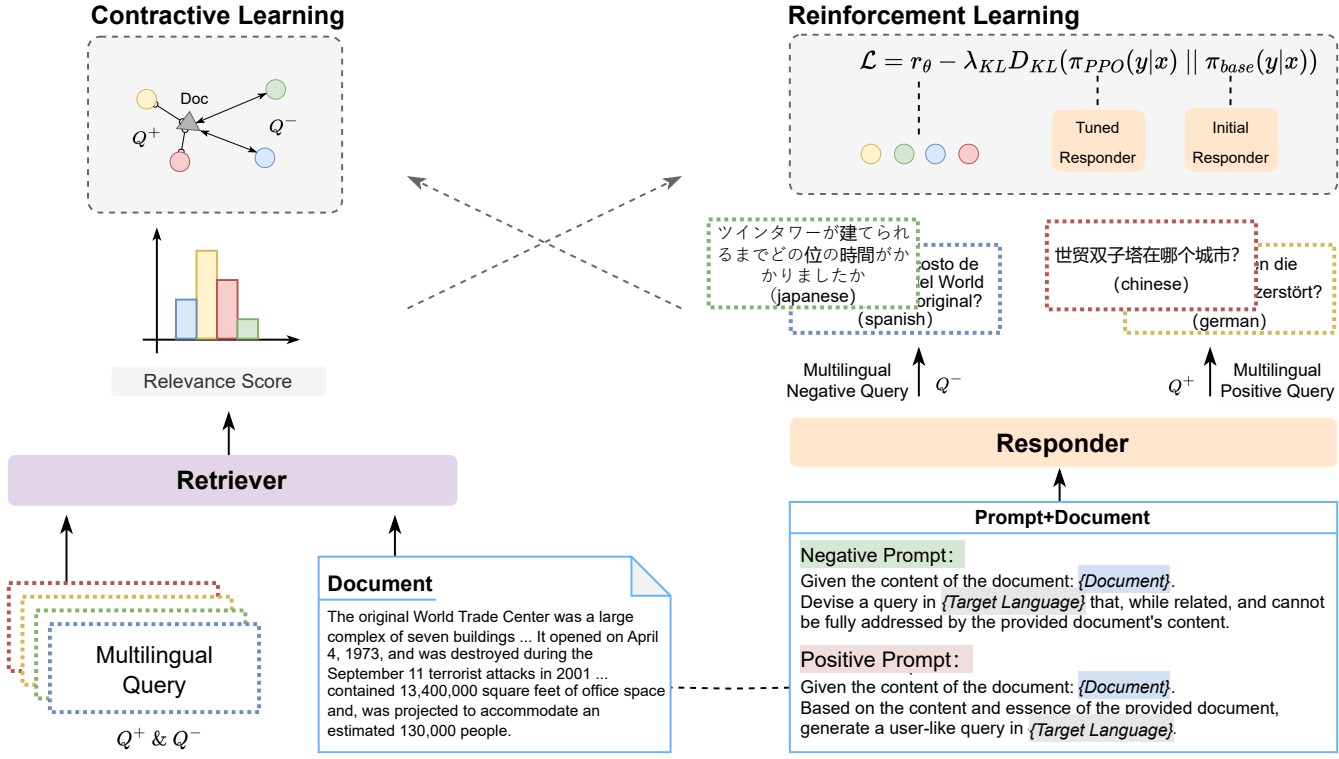

**Figure 1: The overall training framework of Mɪᴍɪᴄ, which is an iterative framework. In one iteration, the _Retriever_ is trained on the synthetic multilingual queries generated by the responder through contrastive learning. In reverse, the _Responder_ is trained under the reward signal scored by the _Retriever_.**

diverse nature of real-world search inquiries and deepens the training experience. With this rich set of queries, the _Retriever_ is primed to navigate a variety of linguistic challenges, ensuring it delivers peak performance in real-world search scenarios.

## 3.2 Retriever Training with Synthetic Queries

To ensure excellence in cross-lingual document retrieval, the _Retriever_ ($R_t$) undergoes intensive training with our crafted synthetic query sets. This rigorous training process amplifies the _Retriever_'s proficiency in discerning relevant documents across a myriad of languages. Specifically, the _Retriever_ learns to gravitate towards positive queries when linked with a pertinent document and, conversely, distances itself from negative queries that are not in alignment with the document's content. This behavior is captured using a contrastive loss, designed such that embeddings of positive queries are drawn nearer to their associated documents in the embedding space, while the embeddings of negative queries are repelled. Mathematically, the contrastive loss $\mathcal{L}_{R_t}$ can be expressed as:

$$\mathcal{L}_{R_t} = \frac{\sum_{q \in Q^+} \text{Rel}\big[R_t(D_y), R_t(q)\big]}{\sum_{q \in Q^+} \text{Rel}\big[R_t(D_y), R_t(q)\big] + \sum_{k \in Q^-} \text{Rel}\big[R_t(D_y), R_t(k)\big]}$$

$$\text{Rel}\big[R_t(D_y), R_t(q)\big] = \text{sigmoid}\Big[\mathbf{W} \cdot (R_t(D_y)||R_t(q)) + \mathbf{b}\Big]$$

$$(1)$$

Here, $\text{Rel}\big[\cdot, \cdot\big]$ represents the semantic relative score, for which we employ a dense linear layer parameterized by $\mathbf{W}, \mathbf{b}$. $R_t(D_y)$ retrieves the embedding representations of the document $D_y$ through the _Retriever_. $[||]$ means the concatenation operation. Note that $R_t(D_y)$ can be more than a single embedding, it can represent all kinds of information we use to represent the document $D_y$. We take $D_y$ as the document representation for simplicity. Through this training approach, the _Retriever_ is finely calibrated to deliver unparalleled performance in cross-lingual document retrieval tasks.

After the fine-tuning process of the _Retriever_, the semantic relevance score between a document $D_y$ and a query $q$ is determined with the score $\text{Rel}\big[R_t(D_y), R_t(q)\big]$. To provide a consistent and interpretable reinforcement signal for the _Responder_ ($R_p$), we rescale the relative score $\text{Rel}\big[R_t(D_y), R_t(q)\big]$ to lie within the interval $[-\delta, \delta]$. The reward score, $\mathcal{R}$, is then calculated as:

$$\mathcal{R}(q, D_y) = 2\delta \times \frac{\text{Rel}\big[R_t(D_y), R_t(q)\big] - \text{min\_rel}}{\text{max\_rel} - \text{min\_rel}} - \delta \quad (2)$$

where max_rel and min_rel represent the maximum and minimum relevance values of the document $D_y$, respectively. This transformation ensures that the reward score spans the spectrum of relevancy between the query and document, assisting the _Responder_ in crafting superior queries by heeding the feedback encapsulated in the reward.

## 3.3 Reinforcing the Responder with Cross-lingual Proximal Policy Optimization

Once the reward signals are derived from the *Retriever*, they become instrumental in steering the training of the *Responder* ($R_p$). In particular, we employ reinforcement learning (RL) techniques to optimize the query generation process of the *Responder* based on these reward signals. Given a document $D_y$, the *Responder* crafts a query $q$, guided by the previously mentioned prompts, and subsequently evaluates its quality by consulting the reward signal. The objective is to maximize the expected reward:

$$\mathcal{J}(\theta) =$$
$$\mathbb{E}_{q \sim \pi_\theta(\cdot|D_y)}\left[r_t(\theta) - \lambda_{KL}D_{KL}\big(\pi_{PPO}(y=q|x=D_y)||\pi_{base}(y=q|x=D_y)\big)\right]$$
$$(3)$$

where $\theta$ are the parameters of the *Responder*, $D_{KL}$ represents the KL-divergency, and $\pi_\theta$ represents the policy of generating a query when provided with a document. $r_t(\theta)$ is the ratio of the current policy to the old policy. We introduce the *Cross-lingual Proximal Policy Optimization (X-PPO)*, a tailored adaptation of the traditional PPO suited for multilingual contexts:

$$\mathcal{L}_{X-PPO,l} = \mathbb{E}\left[\min\left(r_t(\theta)\hat{A}_t, \text{clip}(r_t(\theta), 1-\epsilon_l, 1+\epsilon_l)\hat{A}_t\right)\right] \quad (4)$$

where $\hat{A}_t$ denotes the advantage estimate, which is calculated based on $\mathcal{R}(q, D_y)$. Details about the PPO variables can be found in the following papers [37, 41, 65]. To elucidate the components of this approach:

**Dynamic Clipping Range** $\epsilon_l$: Capturing the unique training trajectories languages might exhibit:

$$\epsilon_l = \epsilon_{base} + \beta \times \text{Var}(\mathcal{L}_{X-PPO,l})$$
$$\text{Var}(\mathcal{L}_{X-PPO,l}) = \frac{1}{n}\sum_{i=1}^{n}(\mathcal{L}_{X-PPO,l}^{(i)} - \overline{\mathcal{L}}_{X-PPO,l})^2 \quad (5)$$

where The $\epsilon_{base}$ represents a base clipping value, and $\beta$ serves as a scaling factor to control the impact of the variance on the clipping range. $\mathcal{L}_{X-PPO,l}^{(i)}$ is the loss at the $i$-th epoch for language $l$ and $\overline{\mathcal{L}}_{X-PPO,l}$ is the mean loss for the last $n$ epochs. The culmination of these elements in X-PPO ensures a nuanced reinforcement learning regime. It uniquely positions MIMIR to adeptly navigate the multifaceted landscape of multilingual information retrieval.

## 3.4 Overall Procedure for MIMIR

To make the overall training procedure much easier to understand, we summarize the training procedure of MIMIR in Algorithm 1.

In the MIMIR framework, the search process is straightforward. When a user provides a query $q$, the *Retriever* scans our multilingual document set $\mathcal{D}$ to find the most $K$ relevant documents. Once these are identified, the *Responder* uses them, along with the user's query, to generate a clear answer with the following prompts:

*"Given the user's query, [User_Query], and the relevant document information, [Document_Content], please formulate a clear and concise answer in the same language as the user's query that effectively addresses the user's question.".*

When plugged into the model, the placeholders *[User_Query]* and *[Document_Content]* would be replaced with the actual content

of the user's query and the selected relevant document, respectively. This approach improved through our training methods, ensures accurate and context-aware responses.

---

**Algorithm 1** Training Procedure for MIMIR

---

**Require:** Unsupervised Multilingual Document set $\mathcal{D}$, Pre-trained *Responder* ($R_p$), Pre-trained *Retriever* ($R_t$)
**Ensure:** Fine-tuned *Responder* ($R_p$), Fine-tuned *Retriever* ($R_t$)
1: **Training:**
2: **while** not converged **do**
3:     **Step 1:** Synthetic Query Generation Using *Responder*
4:         Construct $Q^+$ with Positive Prompt
5:         Construct $Q^-$ with Negative Prompt
6:     **Step 2:** *Retriever* Training with Synthetic Queries
7:         Fine-tune $R_t$ with Eq. 1
8:         Calculate reward signal $\mathcal{R}(q, D_y)$ with Eq. 2
9:     **Step 3:** Fine-tuning the *Responder* with $\mathcal{R}(q, D_y)$
10:        Fine-tune $R_p$ with Eq. 3
11: **end while**
12: **return** *Responder* ($R_p$), *Retriever* ($R_t$)

---

## 4 EXPERIMENTAL SETUP

In our assessment of MIMIR, we focus on two core tasks in the cross-lingual domain: Cross-lingual Information Retrieval (CLIR) and Multilingual Knowledge-Based Question Answering (MKBQA). This dual evaluation is key to understanding the full capacity of MIMIR. With CLIR, we assess the *Retriever*'s skill in finding relevant documents based on synthetic queries. The MKBQA task, in contrast, evaluates the *Responder*'s ability to provide precise answers to user queries. Together, these tasks allow us to comprehensively evaluate both retrieval and response capabilities of MIMIR.

### 4.1 CLIR and MKBQA Settings

**CLIR settings.** Our primary objective in this setting is to tackle a prevalent scenario arising from the abundance of online English data: processing non-English queries against an English document collection. To ensure a meticulous evaluation of cross-lingual retrieval performance in MIMIR, we utilize human translations of a standard query set. This enables us to secure queries in diverse languages. Nevertheless, despite this variation in query translations, the content and language of the retrieval corpus remain consistent. For a balanced comparison with previous methods [18], we've chosen four low-resource languages from distinct linguistic families: Niger-Congo (Swahili), Afro-Asiatic (Somali), Austronesian (Tagalog), and Indo-European (Marathi). We also incorporate three medium to high-resource languages—Finnish, German, and French—to provide a more comprehensive insight into MIMIR's performance.

**MKBQA settings.** Our objective here is to evaluate how effectively MIMIR can generate cross-lingual answers, rather than retrieving documents from a collection. Traditional evaluation strategies, reliant on exact match for retrieval models, are ill-suited for this task which involves comparing the model's output to a reference answer to gauge accuracy. While LLMs typically produce text

paragraphs embedding answers, these might not always mirror precise answers. Often, they present a reformulation of the reference answer. For results that mirror the exact match evaluations with LLMs, we adopt the token overlap recall score for the initial 2000 tokens (R@2kt). In our assessment of Mimir, it is tested across 10 languages, including German, Spanish, French, Italian, Norwegian, Portuguese, Thai, Turkish, Vietnamese, and Chinese, in line with Sorokin et al. [44].

## 4.2 Dataset

**Evaluation data.** Our focus encompasses two distinct tasks: retrieval from English collections using multilingual queries and generating accurate answers in multiple languages with English collections. Accordingly, we formulate three test sets, varying in collection size, relevance distribution, and language configurations.

- **CLEF.** The data derived from the Cross-Language Evaluation Forum (CLEF) campaigns from 2000-2003, were specifically tailored for bilingual ad-hoc retrieval tracks. We pre-process this data following the methods of Huang et al. [18]. Queries are constructed by concatenating the title and description fields from the topic files. Overall, the dataset contains 151 queries from the CLEF C001 – C200 topic, omitting queries without relevant judgments. The English document collection is sourced from the Los Angeles Times corpus, which boasts 113k news articles. For Finnish, German, and French, the queries are provided by the CLEF campaign. For low-resource languages, Bonab et al. [5] supplies Somali and Swahili translations of English queries. Additionally, we enlist bilingual human experts from the Gengo service to translate English queries into Tagalog and Marathi.

- **MKQA.** The MKQA dataset [32] is an exhaustive benchmark tailored to evaluate open-domain question answering (QA) within a multilingual context. Featuring over 10,000 examples, it provides questions in 26 unique languages, ensuring each English question is complemented by 26 high-quality translations. For our evaluation of Mimir, we select 20 out of the available 26 languages, aligning with Sorokin et al. [44]. The dataset's answers, sourced from open-domain passages, can vary in form—from numbers and dates to concise phrases. With its vast linguistic range, the MKQA dataset serves as a crucial tool for assessing the adaptability and precision of QA systems across different languages.

**Supervised warm-up data.** To ensure the stable and consistent performance of Mimir during the iterative process, we utilize multilingual triples from the MS MARCO dataset to warm up both the *Retriever* and *Responder* modules. From this dataset, we randomly select a subset comprising 7 million cross-lingual triples per language, thereby constructing a multilingual training set. As our warm-up strategy for the *Retriever*, we adhere to the fine-tuning methods described in Huang et al. [19]. For the *Responder*, we employ cross-lingual Question-Answer pairs extracted from MS MARCO triples to perform supervised fine-tuning (SFT).

**Unsupervised training data.** The iterative fine-tuning framework within Mimir operates in an unsupervised manner. For this purpose, we source multilingual data from Wikipedia[2]. Notably, our collection only requires English document data, since Mimir is designed to autonomously generate multilingual queries based on the English content. For data extraction, we utilize WikiExtractor[3] on the Wikipedia database backup dump[4]. Following the data pre-processing, we randomly select 50 million English sentences to facilitate the iterative training of Mimir.

## 4.3 Implementation Details

Mimir's architecture is underpinned by two pivotal components: the *Retriever* ($R_t$), initialized using the multilingual pre-trained LaBSE model, and the *Responder* ($R_p$), based on the multilingual instruction-tuned BLOOMZ-7B1 model. In the synthetic query generation phase, the *Responder* is tasked with generating queries in 20 different languages, as detailed in Section 4.1. During the retrieval training phase, our focus was on enhancing the *Retriever*'s efficiency using both synthetic positive and negative query sets. We maintained a consistent sampling of positive and negative queries at $N^+ = 5$ and $N^- = 25$, respectively, to strike a balance between the queries and the document. During the contrastive learning fine-tuning stage for the *Retriever*, we utilized a learning rate of $3 \times 10^{-6}$, processing in batches comprising 4 documents each, yielding an effective batch size of 120. For the *Responder*'s reinforcement learning fine-tuning, we adhered to hyperparameters in line with the PPO framework. For determining the dynamic clipping range, $\beta$ was set at 0.95, and the base clipping range parameter was $\epsilon_{\text{base}} = 0.2$. This phase processed data in batches of 256, employed gradient accumulation, and used a learning rate of $5 \times 10^{-5}$. Convergence was achieved in three epochs. All experiments were conducted using PyTorch, supported by the Huggingface[5] and DeepSpeed-Chat [57] toolkits.

**Evaluation.** To gauge retrieval effectiveness, we draw from established methodologies on the CLEF dataset [2, 11, 16, 18, 19], reporting both the mean average precision (MAP) for the top 100 and the precision for the top 10 (P@10) ranked documents. Statistical significance was ascertained using a two-tailed paired *t*-test with a p-value threshold of 0.05. For assessing generation quality, we followed the methodology from prior work [44] and presented the recall scores for the initial 2000 tokens (R@2kt).

## 4.4 Compared Methods

We compare Mimic with the methods in the following:

- **SMT+BM25**: This approach leverages the Statistical Machine Translation (SMT) method to translate queries. Using the GIZA++ toolkit, a translation table is built for each language pair. The top 10 translations from this table are selected for each query term, which is then used with Galago's weighted #combine operator to generate a translated query. BM25 is then employed to retrieve documents using the translated queries.

- **NMT+BM25**: Utilizing the superiority of Neural Machine Translation (NMT) models over SMT in translation quality,

---

[2]https://www.wikipedia.org/
[3]https://github.com/attardi/wikiextractor
[4]https://dumps.wikimedia.org/
[5]https://huggingface.co/

**Table 1: A comparison of model performance on CLEF benchmark. The highest value is marked with bold text. We have fine-tuned LaBSE using the same supervised data and report the fine-tuned performance.**

| Retrieval Methods | Low Resource Languages | | | | | | | | Medium or High Resource Languages | | | | | |
| --- | --- | --- | --- | --- | --- | --- | --- | --- | --- | --- | --- | --- | --- | --- |
| | Swahili | | Somali | | Tagalog | | Marathi | | Finnish | | German | | French | |
| | MAP | P@10 | MAP | P@10 | MAP | P@10 | MAP | P@10 | MAP | P@10 | MAP | P@10 | MAP | P@10 |
| SMT+BM25 | 0.2271 | 0.2139 | 0.1978 | 0.1832 | 0.1655 | 0.0951 | 0.1047 | 0.0965 | 0.3089 | 0.2810 | 0.3921 | 0.3419 | 0.4052 | 0.3754 |
| NMT+BM25 | 0.2187 | 0.2088 | 0.1448 | 0.1356 | 0.3527 | 0.3202 | 0.1820 | 0.1781 | 0.3742 | 0.3603 | 0.4092 | 0.3595 | 0.4299 | 0.3862 |
| Code-Switch | 0.2420 | 0.2258 | 0.1845 | 0.1682 | 0.3542 | 0.2934 | 0.1573 | 0.1662 | 0.3831 | 0.3403 | 0.4553 | 0.3827 | 0.4589 | 0.3993 |
| Translate-Test | 0.2632 | 0.2537 | 0.2132 | 0.2098 | 0.3816 | 0.3355 | 0.2155 | 0.2246 | 0.4401 | 0.3889 | 0.4795 | **0.4091** | 0.4988 | 0.4234 |
| OPTICAL | 0.3129 | 0.2901 | 0.2477 | 0.2365 | 0.4188 | 0.3623 | 0.2414 | 0.2384 | 0.4228 | 0.3874 | 0.4832 | 0.4067 | 0.4764 | 0.4119 |
| LaBSE | 0.3185 | 0.2998 | 0.2581 | 0.2605 | 0.4207 | 0.3773 | 0.2762 | 0.2505 | **0.4405** | **0.4038** | 0.4874 | 0.4030 | 0.4896 | 0.4090 |
| Mimic | **0.3482** | **0.3269** | **0.3214** | **0.2956** | **0.4498** | **0.3815** | **0.3029** | **0.2841** | 0.4364 | 0.3991 | **0.4912** | 0.4053 | **0.4991** | **0.4307** |

this method first translates the query into English using an NMT model. The translated query is then subjected to the BM25 algorithm for document retrieval.

- **Code-Switch**: This method focuses on data augmentation techniques that enhance training for cross-lingual tasks. Qin et al. [38] introduced a code-switching framework that turns monolingual training data into mixed-language data. Taking this further, Bonab et al. [5] suggested a shuffling algorithm to intersperse and mix the translated terms into the query. The code-switch method is applied to queries in the MS MARCO triples, which is then used to train the ColBERT retrieval model.
- **LaBSE**: The *Retriever* in Mimic draws its initialization from LaBSE [9]. Once trained on the MS MARCO triples, this *Retriever* can be directly run on the CLIR evaluation data in a zero-shot setting.
- **OPTICAL**: The OPTICAL approach by Huang et al. [18] treats the cross-lingual token alignment task as an optimal transport problem. It learns by distilling knowledge from a proficient monolingual retrieval model. Notably, it requires bitext data for the distillation training phase.
- **Translate-Test**: Mirroring the NMT-BM25 method, this approach uses an NMT model to translate the evaluation query into English. Once translated, an English-to-English query-document matching is executed using a trained monolingual neural retrieval model like ColBERT.
- **BLOOMZ-7B1**: Muennighoff et al. [36] offers the BLOOMZ, a publicly accessible multitask model instruction fine-tuned on the BLOOM basis, which is renowned as one of the highly multilingual Llms, having training across 46 languages. The 7.1B model variant of BLOOMZ is used post-warm-up on the MS MARCO dataset for experiments.
- **GPT-3.5-TURBO**: Among the most prominent Llms, GPT-3.5-TURBO is proprietary, harnessing the power of instruction tuning, Reinforcement Learning with Human Feedback, and instruction fine-tuning. For the studies, GPT-3.5-TURBO-0301 is accessed via its official Python API.
- **Sentri**: Sentri, as presented by Sorokin et al. [44], employs a singular encoder for both query and passage retrieval from a multilingual collection. Combined with a cross-lingual

generative reader, it sets new standards in retrieval. Remarkably, it can be extended to over 20 languages using a zero-shot approach.

## 5 EXPERIMENTAL RESULTS

The experimental phase of our study evaluated the retrieval performance of Mimic compared to multiple baseline methodologies. Here, we detail the comparative insights and distinct advantages of Mimic.

### 5.1 Retrieval Performance Improvement in Mimic

The comparative results between the baseline methods and Mimic are documented in Table 1. An overarching observation is the dominance of Mimic across all 7 tested languages. The model, on average, outperforms the strongest of the previously established baselines by an impressive 8.8%. Such robust results can be attributed to the high-quality synthetic queries produced by the *Responder*. Unlike traditional methods, Mimic empowers the *Retriever* with richer supervisory signals across various languages. A pivotal component contributing to this supremacy is the strategic design of negative queries, which discernibly distinguish between answerable and unanswerable questions within a given document. Such intricate supervision is evidently beneficial for retrieval tasks, as corroborated by the improvements in Mimic:

**Mimic performs better on low resource languages.** From Table 1, we find that the overall improvements of Mimic over previous baselines is 2.8% on medium and high resource languages, however, the performance gap further rises to 15.4% on low resource languages. Since the low-resource languages often lack a great amount of labeled data to train a retrieval model, the synthetic queries from Mimic can provide more supervision signals than previous methods. From the results in Table 1, we believe the synthetic queries are of great importance in improving the retrieval performance on low-resource languages.

**Cross-lingual encoder performs better than translation models.** Instead of building a CLIR dataset for model training, the Translate-Test method translates the query to English using an NMT model and then retrieves the document based on a monolingual neural retrieval model. With the help of the NMT model at test time,

**Table 2: The performance of post-retrieval translation methods and large language models on MKQA dataset. We report the R@2kt scores and the best performance is marked with bold font.**

| Models | De | Es | Fr | It | Nl | Pt | Th | Tr | Vi | Zh | Avg. |
|---|---|---|---|---|---|---|---|---|---|---|---|
| *Post-retrieval Translation Methods* | | | | | | | | | | | |
| CORA | 44.6 | 45.3 | 44.8 | 44.2 | 47.3 | 40.8 | 45.0 | 34.8 | 33.9 | 33.5 | 41.4 |
| BM25+MT | 43.9 | 45.3 | 41.7 | 41.1 | 45.2 | 46.4 | 45.9 | 42.7 | 44.3 | 38.2 | 43.5 |
| Bi-Encoder | 50.5 | 48.0 | 48.9 | 41.2 | 48.4 | 48.6 | 46.1 | 45.0 | 48.1 | 46.8 | 47.2 |
| Sentri | 56.5 | 55.9 | 55.1 | 54.3 | 56.3 | 54.8 | 55.3 | 53.0 | 54.4 | 50.2 | 54.6 |
| *Large Language Models* | | | | | | | | | | | |
| BLOOMZ-7B1 | 49.3 | 46.9 | 46.7 | 48.8 | 50.1 | 37.0 | 38.8 | 39.5 | 37.9 | 52.1 | 44.7 |
| GPT-3.5-TURBO | 53.8 | 56.5 | **56.0** | 53.2 | 53.9 | 44.5 | 50.2 | 52.0 | 50.5 | 51.0 | 52.2 |
| Mimic | **57.5** | **57.9** | 54.6 | **54.8** | **59.2** | **56.3** | **55.4** | **55.8** | **56.8** | **53.0** | **55.6** |

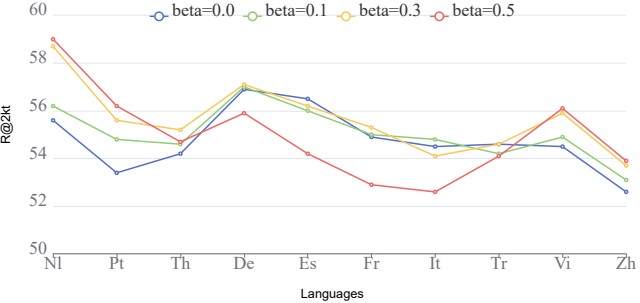

**Figure 2: Analysis about the influence of cross-lingual proximal policy optimization in Mimic. We have set the hyparameters to four different values and report the R@2kt score on ten languages in MKQA. From the result, we set $\beta = 0.3$ for the best performance.**

this pipeline approach can be the strongest baseline in our experiment. From the results in Table 1, we observe that the performance of Mimic on low resource languages is 29.2% percent better than the Translate-Test method, while the Translate-Test method can outperform Mimic by 0.8% on medium and high resource languages. This implies the Translate-Test method is severely influenced by the performance of the NMT model. On low resource languages, it is hard to find an NMT model of good quality, hence the poor performance of NMT model drags down the performance of the overall retrieval performance. As for medium and high-resource languages, obtaining a high-quality NMT model is easy, and with accurate English translation results, retrieving relevant documents is a lot easier, even with traditional statistical methods that can achieve comparable performance. Mimic can train the *Retriever* in an unsupervised manner, hence, it achieves consistent improvements no matter the scale of the datasets in different languages.

## 5.2 More accurate answer generation in Mimic

In Table 2, we have compared Mimic with previous post-hoc translation methods. These methods first retrieved the passages from English Wikipedia, extracted the answer from the top-ranked passage, and translated it with a machine translation model. Compared

with post-hoc translation methods, we find that directly applying Llms on MKQA benchmark cannot match the performance of previous post-hoc translation methods. Even after fine-tuning on the MS MARCO triples, we still observe 9.9 performance gap between BLOOMZ-7B1 and Sentri. The GPT-3.5-TURBO achieves comparable performance when compared with post-hoc translation model, with the performance gap shrinking to 2.4. However, after Mimic fine-tuning, the performance of Llms can exceed the post-hoc translation methods by 1 score in accuracy. This further reveals the effectiveness of the unsupervised fine-tuning paradigm in Mimir. Since both GPT-3.5-TURBO and Mimic have utilized reinforcement learning with human feedback during fine-tuning, this may further prove that reinforcement learning is efficient and useful at achieving cross-lingual consistency on Llms. For the detailed analysis of why Llms can gain better performance than post-hoc translation methods, we leave them in Section 5.5.

## 5.3 Ablation Study

We have reported the performance using only the warm-up model in Table 1 and Table 2. Compared with the warm-up LaBSE model in Table 1, we can find that after using unsupervised iteration fine-tuning in Mimic, the overall performance increases by 5.9% in MAP and 4.9% in P@10. The improvement mainly comes from the low-resource languages. On low resource languages, we find that Mimic gains about 11.7% in MAP and 8.4% in P@10. This proves that Mimic can provide high-quality synthetic queries to help better align the low-resource representations in the *Retriever* space to the high-resource languages. And eventually improves the cross-lingual performance of low-resource languages.

As for the *Responder*, the results are shown in Table 2. Compared with warm-up BLOOMZ-7B1, we find that Mimic can further improve the performance by 10.9 compared with only using MS MARCO triples to fine-tune. This illustrates the effectiveness of Mimic on both *Retriever* and *Responder*. What's more, we conjecture that great improvement may come from reinforcement learning. During reinforcement learning, the *Responder* can learn from a sequence-level feedback signal rather than a previous token-level signal in auto-regressive decoding. This can help improve the quality and relevance of the output. We will dig deeper and explore more in this area in future work.

**Table 3: Cases in real cross-lingual search scenarios. Baseline search methods can not accurately capture cross-lingual named entities and fail to translate under the same topic due to the limitations of the context length. On the contrary, with the large scale of training data, Mimic can alleviate such issues.**

| Case 1 | |
|---|---|
| **Query:** | **Nikujaga** no tsukurikata |
| **Baseline Translation:** | How to make **meat and potatoes**. |
| **Mimic Translation:** | How to make **Nikujaga**. |
| **Mistakes:** | The user might get recipes for various meat and potato dishes, missing out on the specific Japanese dish they were interested in. |

| Case 2 | |
|---|---|
| **Query:** | Técnicas de edición genética más recientes. |
| **Results in English:** | CRISPR-Cas9 has revolutionized genetic editing. It allows for precise DNA modifications. Using **guide RNA**, it targets specific DNA sequences and introduces the desired change. |
| **Baseline Translation:** | CRISPR-Cas9 ha revolucionado la edición genética. Permite modificaciones precisas de ADN. Usando **guía turística de ARN**, se dirige a secuencias específicas de ADN e introduce el cambio deseado. |
| **Mimic Translation:** | CRISPR-Cas9 ha revolucionado la edición genética. Permite modificaciones precisas del ADN. Utilizando **ARN guía**, se dirige a secuencias específicas de ADN e introduce el cambio deseado. |
| **Mistakes:** | "guía turística de ARN" translates to "tourist guide of RNA," a severe error. The translation "guía turística" changes the entire context from a genetic editing scenario to a travel scenario. |

## 5.4 Analysis of the Cross-lingual Proximal Policy Optimization

To better adapt reinforcement learning in the context of cross-lingual sentence learning, we propose cross-lingual proximal policy optimization, which designs different clipping ranges for different languages. To better exploit the impact of the X-PPO, we conduct experiments on different values of the hyper-parameters $\beta$ in the calculation of the dynamic clipping range. The results are shown in Figure 2. $\beta$ determines how sensitive the variation of the loss function can affect the clipping range for each language. Ideally, if the variation in the loss of one language is large, this implies the performance on this language is not fully convergence, hence we tend to broaden the clipping range for this language to make it faster convergence. We have tried different values of $\beta$. When $\beta = 0$, X-PPO will deteriorate to normal PPO, and we find that the performance in different languages varies a lot. We believe that during the pre-training, different languages consume different amounts of data, hence the competence for different languages varies and this cannot be resolved with a constant clipping range. After increasing $\beta$ to 0.3, the performance for those edge languages increases significantly. However, further increasing $\beta$ to 0.5 will not lead to further improvement in the overall performance. This indicates the model may focus on an edgy gradient direction, and lead to general degradation on most languages. Empirically, we set $\beta$ to 0.3 to achieve the best performance in all languages.

## 5.5 Advantages of Mimic than previous translation models

Since the previous translation models suffer from NER issues and cultural context loss issues owing to the restriction of the context length, we conduct some case studies to clearly show that Mimic can help alleviate these issues. We list two cases in Table 3. For case 1, The Japanese query searched for a special dish "Nikujaga", but baseline translation models mistranslated the words into "meat and potatoes", which led to recipes for various meat and potato dishes, missing out on the specific Japanese dishes the user is interested in. While Mimic correctly captures the entity meaning and keeps the special words untranslated for searching. This indicates that after the extensive training data, Mimic has seen numerous entities across different contexts and languages. It can recognize and correctly handle proper nouns, ensuring that they are not inappropriately translated or misrepresented. For case 2, the query is in Spanish asking about the "Latest genetic editing techniques". After searching the corresponding documents, baseline translation models incorrectly translate "guide RNA" to "guía turística de ARN", which even alters the topic of the document. Due to the training on vast and diverse datasets, Mimic has a broader understanding of cultural contexts and a longer accommodation of context length, hence, it correctly translates the word to "ARN guía'. These two cases prove that Mimic can alleviate the problem in baseline translation models. However, Mimic still shows limitations in some query ambiguity and recent culture loss cases, we will focus on building a comprehensive searching framework in future works.

## 6 CONCLUSION

The challenges of CLIR in today's digital age are undeniable. With Mimic, we introduce an innovative solution that leverages Llms to address these challenges head-on. By seamlessly responding in the user's native language and employing a synergistic dual-module architecture, Mimic has demonstrated its edge over existing systems in our evaluations. As the digital landscape evolves, Mimic represents a significant step towards a more inclusive and efficient multilingual information retrieval.

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
