# OpenReview forum: "Query in Your Tongue: Reinforce Large Language Models with Retrievers for Cross-lingual Search Generative Experience"
_ACM.org/TheWebConf/2024/Conference — TheWebConf24 Oral_

### Official Review · Reviewer_8YvP · 2023-11-24

**Novelty:** 6
**Technical Quality:** 5

**Review:**

The authors propose a framework utilizing RL for the problem of Cross-Lingual Information Retrieval (CLIR). Instead of post-retrieval translation based solutions, authors train an end-to-end retrieval and responder combo using RL along with synthetic queries leveraging contrastive learning, which is the biggest contribution of the paper.

Strong Points
1. The paper is fairly easy to follow.
2. Exploration of an end-to-end architecture borrowing GAN's main idea utilizing RL is novel and sounds interesting.
3. The paper provides extensive set of experiments to validate their proposed solution.

Weak Points
1. The number of datasets on which experiments are carried out is not sufficient. Only single dataset for each retrieval and responder comparison is limited to validify their work.
2. The choice of baselines and the decision to separate experiments for retrieval and responder are confusing.
3. I believe the biggest problem is the comparison with baselines does not look fair. Performance gain is most probably due to fine-tuning on the synthetic dataset which is only used for a single baseline in the comparison.
4. Related to the previous point, performance gain especially for retrieval becomes incremental given that there are only 151 queries in the dataset.

**Questions:**

None

**Reviewer Confidence:**

2: The reviewer is willing to defend the evaluation, but it is likely that the reviewer did not understand parts of the paper

**Scope:**

4: The work is relevant to the Web and to the track, and is of broad interest to the community

---

### Official Review · Reviewer_33jp · 2023-11-25

**Novelty:** 5
**Technical Quality:** 5

**Review:**

This paper introduces a new iterative approach to CLIR, based on the use of an LLM to train a better retrieval-side model, and feed its output back to improve the LLM (Fig 1). Specifically, the LLM "responder" would first generate multilingual positive/negative queries of select documents. The generated data is fed into the retriever to improve its effectiveness by using contrastive learning. The same model is used later to provides reward signals for the responder-side generation. This algorithm is detailed in Sec 3.3, with the core design following a policy gradient approach PPO but applied to a cross-lingual setting. This research work is evaluated on two CLIR tasks, one based on CLEF and the other on MKQA dataset. The experimental results (Tables 1 & 2) show that the proposed method can outperform several baselines, including LLMs such as BLOOMZ-7B1 and GPT 3.5 Turbo.

I think on the methodology side the paper has some novelty. Performing contrastive learning over positive/negative query pairs is a very interesting twist (Eq 1). Has this or similar ideas been explored before (say in monolingual IR), and most importantly has this been compared to any doc-pair counterparts already? In your setting a straight-up comparison with doc-pair trained model is perhaps not possible, but might be worth some discussions.

X-PPO has this notion of dynamic clipping range $\epsilon_l$ (Eq 5), which is a main departure from PPO. Is this a novel invention from this work? Have you conducted any experiment to understand its impact (compared to original PPO)?

The literature review is a bit too broad, it covers some ground on the development of CLIR but doesn't really touch on the main influence that inspires this work. A focused treatment on relevant topics such as data augmentation and RL-guided generation would provide better context for the research audience. Sec 2.2 has mentioned some recent research on leveraging LLMs to improve IR. I would love to see this narrative expanded to also cover how generative models are used for data augmentation, and maybe some prior art on reward learning.

Is CLEF still the best benchmark nowadays for assessing CLIR performance? I'm asking this for that the dataset is already 20 years old, would like to know whether more recent options such as CLIRMatrix are also considered.

It seems evident from Tables 1 & 2 that MIMIC (MIMIR?) outperforms all listed baseline methods, but the ablation study is a bit lacking - it's hard to see how much improvement is made by performing query-pair based contrastive learning compared to no data augmentation at all. I would appreciate the experiments more if there were some comparison between zero-shot LLMs (no RL) and the proposed method.

Some minor comments
- The PPO training loop (Step 3, similar to [41]) should also be covered in Algorithm 1.
- In Sec 4.4, please provide some references for the MT components in SMT+BM25 and NMT+BM25.
- Table 1 should include a retriever only model (BLOOMZ-7B1?) as the baseline

Pros
- Proposing a novel iterative approach to CLIR, employing an LLM to enhance the retrieval-side model through contrastive learning with positive/negative query pairs.
- Empirical evidence of improvement over prominent LLMs such as BLOOMZ-7B1 and GPT 3.5 Turbo.

Cons
- Lack of a focused discussion on the main influence inspiring the work.
- Lack of suitable controls in the experiment to tease apart the influence of improvement on individual components

**Questions:**

- Is the warm-up step important to CLIR systems?
- Is the BLOOMZ-7B1 run zero-shot or fine-tuned?

**Reviewer Confidence:**

3: The reviewer is confident but not certain that the evaluation is correct

**Scope:**

4: The work is relevant to the Web and to the track, and is of broad interest to the community

---

### Official Review · Reviewer_mj6W · 2023-12-01

**Novelty:** 5
**Technical Quality:** 5

**Review:**

This paper focuses on improving cross-lingual retrieval and cross-lingual QA. The authors propose to generate synthetic cross-lingual queries using an LLM, creating training data for the retriever. The retriever than in-turn provides reward signals for the generator. The authors evaluated both the retrieval performance and the QA performance, and show that the proposed framework can improve both tasks.

Pros:
- This LLM based framework simplifies traditional translation + heuristic based pipeline. I believe there are much headroom for cross-lingual retrieval and QA tasks that can be addresses with framework.
- Generating negative queries is novel, since most existing work on dense retrieval rely on negative passages.
- Good experimental results.
- Paper is clear written and well-organized.

Cons:
- It is unclear to me why the RL can improve the generator on QA tasks, since the reward is on query generation, not answer generation. I hope the authors can better explain the intuition for RL.
-  Most retrieval baselines used in the paper are BM25, but there are several multi-lingual/cross-lingual dense retrievers from existing work, e.g., mDPR (" Mr. TyDi: A Multilingual Benchmark for Dense Retrieval" Zhang et al, 2021) and mContriever ("Unsupervised Dense Information Retrieval with Contrastive Learning" Izacard et al, 2022). They should be discussed in related work and added to baselines.  Comparing to these more recent baselines would justify if the synthetic query generation is necessary for training the retriever.

**Questions:**

Can you address my 2 concerns in the "Cons" section?

**Reviewer Confidence:**

3: The reviewer is confident but not certain that the evaluation is correct

**Scope:**

4: The work is relevant to the Web and to the track, and is of broad interest to the community

---

### Official Review · Reviewer_5K8Q · 2023-12-01

**Novelty:** 5
**Technical Quality:** 5

**Review:**

The paper presents an approach for cross-lingual generative retrieval based on reinforcement learning, retriever training with synthetic queries and query generation based on reward signals from signals. The motivation is clearly stated and the evaluation is done accordingly. Generative IR is a trendy topic while cross-lingual aspects seem to be less investigated despite the multilingual capacity of LLMs. The idea of combining reinforcement learning, synthetic queries cross-lingual proximal policy optimization seems to be novel. The proposal is particularly efficient for low-resource languages but less performing on medium and high-resource languages.

Limitations:
- The results are reported on a single dataset per aspect (retrieval/QA). The obtained conclusions might be biased to the design of a particular dataset.
- The limitations of the SOTA mentioned in L96-99 (NER issues and cultural topic context loss) and the advantage of Mimic are shown on 2 examples
- The quality of the generated positive and negative queries is not evaluated directly

Minors:
- the paper has too many self-promoting claims
- L376: the sentence seems to be unfinished
- L670: "in Table 1. An overarching observation is the dominance of Mimic across all 7 tested languages" does not reflect the Table 1 content correctly
- L596: Which NMT method?
- L732: it is not clear why Translate-Test is called the "strongest" baseline. It does not seem to be the strongest baseline for low-resource languages
- Fig.2: Why did the authors decide to present languages on X and beta values as trends and not vice-versa?
- L120-121: the claim is not evident

**Questions:**

- Section 5.5 does not provide enough evidence about the advantages of Mimic over the previous translation models. Only 2 examples are analyzed. Why these 2 examples are representative?
- The results of Table 2 are very interesting. Might this high performance of posthoc translation of top-ranked passages from English Wikipedia be explained by the design of the MKQA dataset as the raw answers in this dataset were searched on the web and linked to WikiData? Might it occur that the MKQA dataset contains a lot of answers very close to Wikipedia passages?
- The results are reported based on a single dataset per aspect  (retrieval/QA). How do the authors ensure that the obtained results are not biased to a particular dataset design?
- Was the MKQA dataset used for pre-training the LLMs from the baselines or the proposed method?
- It would be interesting to investigate the quality of the positive and negative query generation based on the given prompts.

**Reviewer Confidence:**

3: The reviewer is confident but not certain that the evaluation is correct

**Scope:**

4: The work is relevant to the Web and to the track, and is of broad interest to the community

---

### Official Review · Reviewer_nHcN · 2023-12-01

**Novelty:** 5
**Technical Quality:** 5

**Review:**

The paper addresses the critical role of search engines in facilitating information access within the contemporary digital landscape, emphasizing the challenges faced in Cross-Lingual Information Retrieval (CLIR). While previous efforts have aimed to enhance CLIR through methods involving Neural Machine Translation models and cross-lingual representation, the paper rightly points out persistent issues such as misplaced named entities and lost cultural context in non-native language queries.

This work introduces a novel approach by proposing the utilization of a retriever to unsupervisedly provide reward signals, guiding the optimization of large models to generate more relevant queries. This aspect is commendable and demonstrates innovation in the field.

I am intrigued by the role of positive and negative prompts in training the retriever. Could the authors consider a more streamlined approach where only positive prompts are used to generate queries, treating them as positive queries, while utilizing other queries in the batch as negative queries for training the retriever? This method seems more elegant and aligns with the principle of Occam's Razor. This streamlined approach could potentially simplify the training process while maintaining effectiveness. I recommend the authors explore and discuss the feasibility and potential advantages of this approach in their work.

**Questions:**

1. Could you elaborate on the decision-making process behind choosing specific positive and negative prompts for training the retriever? How do you ensure the representativeness of these prompts in capturing relevant and non-relevant information?

2. The proposed approach involves using both positive and negative prompts for training the retriever. Have you explored or considered an approach where only positive prompts are used, treating them as positive queries, while using other queries in the batch as negative queries?

3. How sensitive is MIMIR's performance to changes in the size of the retriever and responder modules? Could you conducte experiments to analyze the impact of varying model sizes on efficiency and effectiveness?

**Reviewer Confidence:**

4: The reviewer is certain that the evaluation is correct and very familiar with the relevant literature

**Scope:**

4: The work is relevant to the Web and to the track, and is of broad interest to the community

---

### Decision · Program_Chairs · 2024-01-22

**Decision:**

Accept (Oral)

**Comment:**

This paper presents an approach for cross-lingual generative retrieval based on reinforcement learning, retriever training with synthetic queries and query generation based on reward signals from signals

 The paper was reviewed by five reviewers. The paper has clearly some merits. All reviewers agree on the technical quality and novely of the papers, but they also raise some comments still requirint a proper explanation. Please clarify this point in the camera-ready copy.